# Intelligent Polymers, Fibers and Applications

**DOI:** 10.3390/polym13091427

**Published:** 2021-04-28

**Authors:** Li Jingcheng, Vundrala Sumedha Reddy, Wanasinghe A. D. M. Jayathilaka, Amutha Chinnappan, Seeram Ramakrishna, Rituparna Ghosh

**Affiliations:** Department of Mechanical Engineering, Centre for Nanotechnology & Sustainability, National University of Singapore, Singapore 117574, Singapore; jingcheng@u.nus.edu (L.J.); sumedha@u.nus.edu (V.S.R.); dumith@u.nus.edu (W.A.D.M.J.); mpecam@nus.edu.sg (A.C.)

**Keywords:** intelligent materials, stimuli-responsive, polymers

## Abstract

Intelligent materials, also known as smart materials, are capable of reacting to various external stimuli or environmental changes by rearranging their structure at a molecular level and adapting functionality accordingly. The initial concept of the intelligence of a material originated from the natural biological system, following the sensing–reacting–learning mechanism. The dynamic and adaptive nature, along with the immediate responsiveness, of the polymer- and fiber-based smart materials have increased their global demand in both academia and industry. In this manuscript, the most recent progress in smart materials with various features is reviewed with a focus on their applications in diverse fields. Moreover, their performance and working mechanisms, based on different physical, chemical and biological stimuli, such as temperature, electric and magnetic field, deformation, pH and enzymes, are summarized. Finally, the study is concluded by highlighting the existing challenges and future opportunities in the field of intelligent materials.

## 1. Introduction

The unique properties of polymers have long been gaining attention and investigation from both academia and industry. The characteristics of different polymers rely on how the long chains of the molecules repeat themselves and bond with each other [1]. The inherent structure, along with the way molecules arrange themselves and cross-link, aids materials’ responsiveness to trivial environmental changes [2,3,4]. Intelligent polymers, as the word ’intelligent’ implies, are the polymers that are responsive to single or multiple stimuli, which could be either chemical or physical [2,5].

It is essential at the beginning of this review to define what an intelligent polymer is. To clarify the term “intelligent” with more accurate standards, we look back at the origin of it, which is borrowed from science-fiction literature as the opposite of those that are obtuse to environmental changes, that have no such ability to make choices and that cannot respond or adapt to comprehend complex situations or learn from the past. As well-established terminology in human intelligence and artificial intelligence [6,7], it is the intellectual power that gives human cognitive capabilities along with self-awareness to reason and understand, think and resolve, innovate and design, plan and predict, communicate and interact with each other. In a word, human intelligence can be seen as the ability to define, solve and learn from problems in various scenarios [8,9,10]. Similarly, from the deduction of parallel interpretation, the “intelligence” of a material could be considered to be the capability of property changes responsive to environmental stimuli with corresponding molecular-level structural rearrangement [11,12,13]. These external stimuli, as seen in Figure 1a, could be referring to changes in light intensity, temperature, pH, electricity or magnetic field, mechanical deformation or pressure, biological stimuli, etc. [14,15,16]. The prototype of an intelligent polymer could be defined as a material that comprehends experiences, is self-aware and responds purposefully. Such ability to be aware of environmental changes allows intelligent polymers to adapt to ensure future improved behaviors in similar situations and in certain applications [17,18,19,20]. On the other hand, some researchers have also worked on implementing artificial-intelligence methodologies, like machine learning (ML), into the development of polymers, such as the ML model for polymer swelling in liquids [21], the prediction of point defects in materials [22] and sustainable material synthesis [23].

Keeping an eye on the increasing demands in developing and promoting smart materials in various types of applications in both academia and industry, polymer- and fiber-based materials are of the most interest [24]. With Figure 1b illustrating the trends of major progress and publications in intelligent polymers, the increasing amount of research works in several directions. Figure 2 shows the historical tendency of major engineering material systems to transition from structural to functional with a prospective future of low-carbon sustainable intelligent materials [25]. Major preoccupations have seen materials starting from Stone Age primitive materials and the rudimentary use of chemistry to treat natural rubber and metals, followed by electrochemistry in the last century, with numerous materials developed or discovered in between [26,27]. Nowadays, the wide use of polymers and ceramics makes industrial and daily-life products feasible, affordable and suitable for mass production [28,29,30]. Piezoelectric, semiconducting and thermoelectric materials are seen in many state-of-the-art applications [31,32,33,34] and intelligent devices based on their unique characteristics. Nevertheless, these materials still hold a certain level of limitations regarding the degree of intelligence, as most of them lack certain functionalities such as self-control, decision-making, self-learning or ease of recycling, which are critical to the demanding sophisticated modern way of life [25].

In this review, features of polymer- and fiber-based materials are discussed, followed by applications in various fields and comparisons and summaries of the properties, characteristics and mechanisms of intelligent materials from a perspective of functionalities due to triggering stimuli, which range from physical (temperature, electric and magnetic field and deformation), chemical (pH) and biological (enzyme) basis. Section 2 focuses on the basic working mechanisms of several widely used intelligent polymers, followed by their respective characteristics and comparisons of performance in various applications. After investigations on shape-memory polymers (SMPs) in Section 2.1 and self-healing polymers in Section 2.2, Section 2.3 focuses on multiresponsive polymers that are magnetic-, humidity- and pH-responsive. Section 3 gives a brief review on the following developed fields of polymer applications: drug-delivery systems, smart textiles and polymer-based healthcare wearables. Section 4 summarizes the discussion on intelligent polymers, points out the current research gaps and provides possible future research directions on intelligent polymers.

## 2. Classifications and Underlying Mechanisms of Intelligent Polymers

From the level of macroscopic consideration, the adaptive performances of intelligent materials are achieved by the autonomous behaviors of molecules or atoms at a nanoscopic level [35]. When stimuli are introduced, aggregation, rearrangement and directional movement occur among the molecules and atoms [36,37]. To perform an optimal response—either in reflection of changes in shape or color or in generated electrical signals—to the stimuli, which could be changes in electricity, magnetism, heat, humidity, stress, light intensity, nuclear radiation level, introduced chemicals, etc., the adaptiveness of an intelligent material is shown at both a micro and macro level [14,38]. To obtain an overall understanding of intelligent polymers, in this section, we present the fundamentals of intelligent polymers. Specifically, polymer-based materials and fiber-based materials are introduced in detail from their characteristics and mechanisms to their use-case scenarios, which play important and inevitable roles in modern society. By introducing the working mechanisms of smart polymers utilized in three major fields (shape-memory, self-healing and responsive polymers), the underlying principles of stimuli-responsive features have been discussed. As clarified in the introduction, intelligent polymers hold various working mechanisms for multiple specialized applications, whilst possess memorizing, learning and reacting capabilities from a molecule level.

### 2.1. Shape-Memory Intelligent Polymers

SMPs are polymeric materials that are able to transit between different predefined shapes when external stimuli are introduced [38]. They offer a wide range of applications, both currently and in the immediate future, in biomedical, intelligent devices, medicine, aerospace, photonics, manufacturing, textiles and household goods, as well as microstructure printing for anticounterfeiting [39,40,41,42]. Two approaches are mainly taken, either the viscoelastic approach or the phase transition approach [43], to analyze and model the behavior of SMPs. Significant progress has been made in optimizing and broadening the transforming behaviors of shape-memory polymers to not limit them with bare dual shape-memory transitions [44,45].

Among SMPs with various working principles, thermoresponsive SMPs are most studied [46,47,48,49]. A typical thermoresponsive SMP working cycle [38] is illustrated in Figure 3a. When the surrounding environment is at a lower temperature, the memorized stage of a thermoresponsive SMP has a less-ordered configuration [50]. In shape A, the molecular connections are relaxed and stable with weak thermal motion; the switching segments are elongated and fixed in the polymer. When heat is applied, the polymer shows viscoelastic characteristics as the molecular motion becomes active, chain orientations are switched and net-points are dislocated, leading to a new set of interactions between SMP chains [51,52]. For example, Ansari et al. [53] discussed the angle recovery of SMP under different temperatures. To fabricate complex shape-memory polymer geometry, Zhang et al. [54] reported a material and process concept to allow the fine control of SMP fabrication during a short period of time—30 s—of 4D printing, of which time is the fourth dimension, as depicted in Figure 3c. The idea of 4D printing [55,56] was introduced to enhance the performance of SMPs by improving durability and achieving more complicated shape-memory behavior compared to 3D printing. Additionally, with the help of digital light modulation, the spatial-temporal tuning of the material properties has been achieved [54,57]. In Figure 3b,d, working mechanisms of SMPs under stimuli triggers are illustrated. Bonding in metal retains strength retainability over covalent bonds, while dynamic reversible molecular bonds in polymers serve as a complementary switchable property, in addition to the bonds present in metals that are observed in metallopolymers. Combining the mechanical features of both metal and polymer together in the design of metallopolymer shape-memory material, a unique way to control metal ion diffusion in the linear polymer network [58] was proposed by Yang et al., as in Figure 3b. By doing so, the overall network yields a gradient plasticity. By using this concept, the linear polymer network was synthesized with terpyridine-containing acrylate (Tpy-A), methyl methacrylate (MMA) and butyl acrylate (BA) in a *N*,*N*-dimethylformamide (DMF) solution first, then added to metal salt to obtain the gel form. During characterization, they compared stress relaxation for samples doped with different metal ions, namely Ni, Fe and Zn. As in Figure 3d, another major domain of interest is light-responsive SMPs. Other than the reversible transitions mentioned above, it is possible for the shape-memory effect to be triggered under other circumstances, such as changes in light intensity, humidity and electric fields [59]. For instance, with additional iron oxide added, the SMP can be triggered by UV to present magnetically guidable features [57]. Table 1 lists the performance of polymer-based SMPs with various material combination and their features. Versatile manufacturing methods indicate the feasibility of generating on-demand SMPs for different applications with various composite materials added. A high shape-recovery rate and biocompatibility made them suitable for wearable electronics and health monitoring devices.

Another essential emerging class of SMPs is vitrimers, or covalent adaptable networks (CANs). Such thermosetting polymers are biobased, solvent-resistant and recyclable, thus offering various functionalities in industrial applications, such as protective coatings, reinforced or adhesive composite materials and biomedical devices [66,67]. Vitrimers with dynamic covalent bonds based on associate mechanisms are able to maintain cross-linked networks when heat is applied [68] and are consequently able to build controlled multishape memory vitrimer by hot-pressing [67], as Pei and colleagues reported. With optimized synthesis methods to further investigate, such as introducing 3D-printable composites [66], controlling the tensile strength property by adjusting the molar ratio of crosslinking functional groups [69], improving the mechanical properties and shape fixity to 98% and above by bringing in graphene composites [70], vitrimers have the potential to be applied in cases requiring reconfigurable, self-healing and self-welding features.

To conclude, polymers sensitive to temperature changes can be divided into two categories based on their phase transition behavior. When the temperature is raised, polymers present upper critical solution temperatures (UCST) and undergo transition from biphasic to monophasic [71,72], while those in transit from monophasic to biphasic are considered to be low critical solution temperature polymers (LCST), leading to a transition of hydrophilic to hydrophobic behavior in the polymer, the latter of which has more investigation than the former. The reason radical change occurs in the solubility of thermoresponsive polymers when the temperature changes is that a miscibility gap is observed in the phase diagram [72,73]. Besides commonly used LCST polymers such as poly(N-substituted acrylamide) and poly(vinylamide), and including poly(oligoethylene glycol (meth)acrylate) families, there are also others that present similar temperature-responsive behavior when macromolecular hydrophilic and hydrophobic balance is satisfied [74]. This feature could be useful in drug delivery applications [72,73,74,75], designing thermoreactive self-folding materials [76], the controlled permeability of fiber coatings [77] and membranes [78].

### 2.2. Self-Healing Polymers

Mechanical stimuli, such as stress, strain and twist, cause changes in the form of polymers, and some polymers show recovering properties to transfer back to their previous defined shapes [79,80,81]. Compared with metals and ceramics, these polymers are easier to process at lower costs [82,83]. The self-healing ability of a polymer is obtained either extrinsically (from the polymer structure), or intrinsically (from the material molecular chain) [84]. Extrinsic healing comprises the majority of cases, in which healing compounds are stored in capsules or nanoparticles isolated from the polymer matrix. Otherwise, intrinsic self-healing polymers utilize the mobility of molecule chains by rearranging their configuration. Either way, the original functions, such as electrical conductivity, integrity of the surface structure or other mechanical properties of the material, are restored [85], which is where the term “self-healing” comes from.

Since most studied self-healing polymers are in gel form [84], polymer hydrogels are vital in the process of manufacturing modern biomaterials, as their richness in hydration and similar three-dimensional structure render themselves compatible with natural tissue [86,87]. Stretchable, switchable and elastic, hydrogels are hydrophilic with cross-linking structural networks [88]. However, despite these outstanding features, hydrogels’ applications still present a certain level of limitation due to their weakness in bonding, inability to be controlled, difficulty to actuate and complexities in design-responsive polymers [89]. When single or multiple stimuli are applied to the material, macroscopic responses, such as swelling/collapse or solution–gel transitions, are induced, depending on the physical state of the chains [3].

Two prerequisites are considered for the material to recover from the mechanical deformation: switching transition between different phases should be reversible, supported by a stable polymer network responsible for a stable original shape. After mechanical deformation is stimulated, shape-memory polymers can fix the temporary structure via various transitions, such as liquid crystal anisotropic transition, molecule cross-linking, crystallization melting or supramolecular association/disassociation [1,90]. Normally, shape-memory polymers show recoverability from strains, but due to poor mechanical properties, less satisfying recovery from stresses is observed [3]. Therefore, increasing attention and investigations aim to develop reinforced properties of shape-memory composites.

In other cases, polymers play important roles in adding to the damage tolerance in deformable electronic devices, making the self-healing of mechanically defected electrical metals possible. Polymer composites serving as a passivation film was reported by Kunmo et al. [91]; where metal conductors are damaged, filler released from liquid metal capsules self-heals the injured site and provides recovered conductivity, as in Figure 4a. In this case, liquid metal capsules are wrapped with a polymer matrix of poly(urethane acrylate) (PUA) to heal the Au contact in the device. After cutting the passivation film, the power-conversion efficiency of the electronic device is well-maintained with a reduction of only around 3%. Another study proposed by Luo et al. [92] involves embedded silver nanowires to the self-healing polymer matrix to diversify its conductivity with respect to monitoring the stages of healing. Figure 4b,c, depicts the self-healing behavior of polymer gels. A comparison of the performance of selected polymers responding to mechanical deformation is listed in Table 2. As shown, polymer-based shape-memory materials hold advantages in almost full-ratio recoverability, extensive durability and prominent stain resistance, compared to generally less than 50% in shape-memory alloys [93,94].

### 2.3. Responsive Polymers Triggered by Other Stimuli

Although mechanical- and thermal-responsive polymers are gaining a significant amount of attention among researchers, as mentioned in previous sections, other types of responsive polymers are also studied, such as those responsive to magnetic fields, humidity fluctuations or pH changes. Most likely, these different responsiveness are combined to achieve the maximized performance of designed functionalities.

Magnetic-resistant polymer gels and elastomers, for example, are composites based on magnetic nanoparticles scattered in a high-elastic polymeric matrix [98], so that their behaviors can be controlled spatiotemporally via external magnetic fields [99,100]. For those composites, adding magnetic nanoparticles leads to added magnetic features, so that materials can interact with magnetic fields [101]. As a result, the magnetic field easily deforms the polymer matrix without noise, heat or fatigue, making it suitable for the preparation of sensors, micromachines, energy transducers, controlled distribution systems and environmental and biomedical applications [102,103,104]. Although the uncontrollable manner of the way magnetic nanoparticles scattered in the gel hindered the promotion of its applications, this issue has been solved by forming cross-linking hydrogel nodes [98,105]. For instance, as shown in Figure 5a, Perera et al. [106] reported one novel type of drug carrier for controlled drug release due to the magnetic sensitivity of biocompatible microfibers. Polyvinyl alcohol incorporated with Fe_3_O_4_ MNP (magnetic nanoparticles) is used to fabricate biocompatible magnetically actuated microfibers. Such magnetically actuated composite microfibers are triggered to release active pharmaceutical ingredients by remote control, which makes accurate and secure pathways of drug delivery possible. The chain behavior in magnetic gel before and after introducing a magnetic field is shown in Figure 5c.

With a combination of novel techniques such as 3D printing, future magneto-responsive polymers can be functionalized in an economic customizable shape [107]. Synthesized 3D-printer ink makes dispersing magnetic particles and flakes in a gel matrix feasible. The obtained magneto-responsive ink can be further used in a wide range of printed applications such as medical devices, micro robotics or magnetic control [108]. The ability to respond to noninvasive, external magnetic fields contributes to the applications of adaptive soft materials [109].

**Figure 5 polymers-13-01427-f005:**
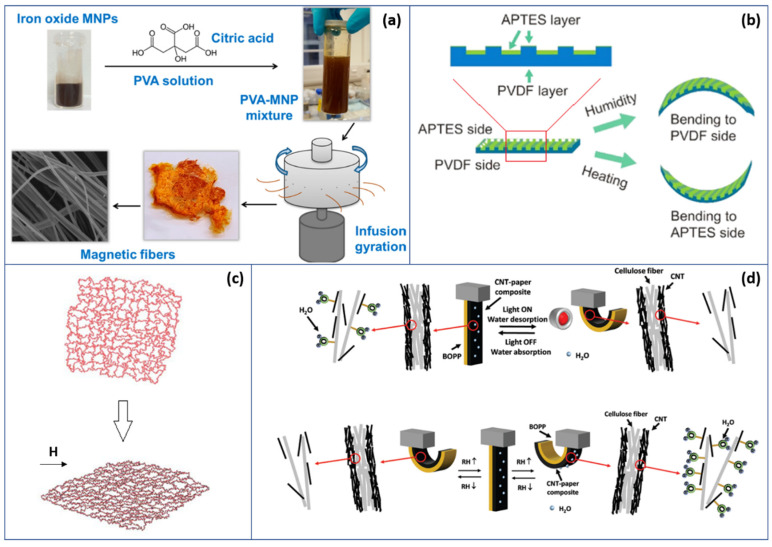
Polymers responding to various stimuli: (**a**) magnetic fiber for controlled drug delivery [106]; (**b**) multiresponsive kinematics and robotics in polymer films with a designed shape-programmable surface pattern (Reprinted with permission from Liang, S.; Qiu, X.; Yuan, J.; Huang, W.; Du, X.; Zhang, L. Multiresponsive Kinematics and Robotics of Surface-Patterned Polymer Film. *ACS Appl. Mater. Interfaces* **2018**, *10* (22), 19123–19132. Copyright American Chemical Society) [110]; (**c**) comparison of chain behavior in magnetic gel before (top) and after (bottom) a magnetic field is applied [105]; (**d**) humidity and light-driven actuating mechanisms of CNT-paper/BOPP composites [111].

Humidity sensing is also another well-studied field of intelligent polymers. A variety of sensing applications is also another well-studied field of intelligent polymers. A variety of applications have been developed taking advantage of moist characteristics of polymers, including humidity sensors, actuators and membranes (which have various applications such as biomimetic fibrous fabrics [110,112,113]). Response to humidity means the hydrophilic polymer should have porous networks to form the fiber structure [110] to let water molecules go through. When ambient humidity is raised, polymers are to absorb those water molecules and desorb in low-moisture circumstances. Depending on working mechanisms, humidity sensors can be divided into two categories: namely, capacitive sensors and resistive sensors [110,114]. As capacitance or resistance values change in response to surrounding humidity changes, such as from adsorbing or releasing water molecules, a signal response from the electric circuit is generated [16].

The multiresponsiveness of the polymer allows its applications to encompass various combined features. In these cases, polymers are normally designed to be responsive to not only humidity but also to other stimuli such as heat, light or chemical vapors. Recently, actuating applications have been investigated, wherein using polymers to convert chemical or physical energy into mechanical energy enables them to serve as actuators. As in Figure 5d, Zhou et al. propose a multistimuli-driven actuator [111] that can sense both light and humidity changes, where the hygroexpansion effect of the CNT-paper composite corresponds to humidity sensitivity, and the mismatch of the CTE (coefficient of thermal expansion) between the BOPP (biaxially oriented polypropylene) layer and CNT-paper composite layer corresponds to light sensitivity. Such actuators can be used in biomimetic applications, such as artificial biomimetic muscle or skin [115]. Another typical solution to implementing multiresponsiveness in polymers, as Chakraborty et al. reported, is to introduce and synthesize functional groups, such as carboxylic acid groups, in the design of a multifunctional Pt(II)-based metallo-supramolecular polymer (polyPtC). In this case, when external stimuli are introduced, Pt-Pt [116] interaction, which is noncovalent, may lead to dynamic changes at a macroscopic level and end with irreversible results. Although they exhibit multifunctional behavior, such metallo-supramolecular polymers are inappropriate to be considered as “intelligent”.

On the other hand, sensing applications are also performed. By varying microchannel alignment, in Figure 5b, Liang et al. [110] managed to develop reversibly responsive film depending on the divergence of absorption abilities between patterned poly(vinylidene fluoride) (PVDF) and 3-aminopropyltriethoxysilane (APTES) films of humidity. With the design of microchannels and various aligned angles, the cut strips can perform heterologous coiling motility. Table 3 includes a selected comparison of multiple stimuli-driven polymer applications.

## 3. Overview of Intelligent Polymer-Based Applications in Multiple Fields

Although smart polymers are widely used in the biomedical field, aiming at develop new therapies for disease treatment or cautiously designed medical devices that react to surrounding tissues or external stimuli, this chapter gives special emphasis on the relevant applications of smart polymers and their future trends within the field of electrical and material science and mechanical engineering based on three years of recent research. Moreover, with functional molecules incorporated into the chemical structure, most polymers are easily functionalized to achieve our desired properties through prepolymerization or postpolymerization. In other words, the potential for designing new materials or modifying them to meet a wide range of application-need-based criteria standards or specific requirements is unlimited and worth exploring. Table 3 in the previous section details a performance comparison of multiple stimuli-driven polymer applications, listed considering various fabrication methods, materials, working mechanisms and features. For example, thermal-responsive polymers can be applied even in a wide range of applications such as data storage [120]. In this chapter, an attempt is made to discuss the other three major fields of applications. To do so, we briefly reviewed polymers used in drug delivery systems, smart textiles and polymer-based membranes and fibers. Trigger-dependent sustained drug-release systems are presented in Figure 6a,b.

Polymer-based nanocarriers enable controlled drug delivery at the right place and right time. The early stages of research on dendrimer-based drug carriers [124] derived the structural designs of polymer drug carriers that are now investigated and developed. Acting in a crucial role in the production of active and selective therapeutic applications [125,126], the further understanding of molecular biology and ways to synthesize newer, multistimuli-responsive polymers has resulted in more efficient, precise and customized therapies. Some polymers can react to environmental pH fluctuations to protonate or deprotonate correspondingly [123,127,128] as Figure 6c shows. When the pH condition of their surroundings changes, these polymers change their conformation, surface behavior and solubility by acquiring or losing ionizable basic groups or acid groups, therefore tuning the solubility of the polymeric nanoparticles [129], fabricating ion-stabilized membranes [130] or designing core-shell nanoparticles [131].

Such characteristics are the origin of how the pH-sensitive release of drug-delivery carriers works. As a promising approach especially in cancer treatments, localized drug-delivery carriers made by polymers decrease the toxicity of the drug and, via site-specific release, enhance the effect of the active compounds [73,106,118,132]. For example, intelligent biocompatible cellulose nanofibers (CNF) developed by He et al. [133] enable sustained drug release via the pH responsiveness of grafted polyethylenimine (PEI). The generated CNF-PEI displayed a rapid response to pH, reflected by its wettability, converting to hydrophobicity when surrounding pH changed from acidic to alkaline and returning to hydrophilicity when pH conditions reversed back. Entering a new generation of enhanced widely used drug-delivery systems for humans has resulted in safety issues and research gaps requiring further investigation. Current research interests includes minimizing the degradation of pharmaceutically active ingredients during the in vivo transporting process [134] and optimizing the formulation methodology of nanoparticles with better release precision [135], as shown in Figure 7.

For example, for site-specific drug delivery, Tiwari et al. developed a tailorable fibrous platform [118] that can release rationally controlled drugs in an environment with pH 5.5, triggered by pH and near-infrared radiation (NIR). Particularly when combined with the NIR-responsive nature of polymers, photothermal therapy has become an alternative therapeutic solution to cancer and has noninvasive benefits [111,118,137]. Other cases of combining polymers with magnetic fields or magnetic nanoparticles have been reported. For instance, Perera and Zhang proposed a way that the remote-controlled release of a drug can be achieved [106]. In their study, Fe_3_O_4_ nanoparticles are used to generate fibers through infusion gyration. As Deshpande et al. reported [121], a drug-delivery system of doxorubicin with polymeric shell nanoparticles and a gold core was developed. The release depends on the thermo- and radiofrequency-responsive features of biocompatible poly(N-isopropylacrylamide) (pNIPAm)-based polymer shells.

Textiles have also experienced great improvements through the incorporation of different kinds of smart polymers to their formulation. Temperature, pH, moisture and light were responsible for the variable aesthetic appeal, smart controlled drug release, wound monitoring and smart wetting [138] properties of new textiles. In addition, textiles that provide safety against significant changes in weather environments and textiles with medicinal properties has also been accomplished through smart polymers [139,140,141]. Other than smart polymers, other materials are also making smart textiles accessible and feasible. In Figure 7a, Ma et al. reported a novel type of yarn-shaped fiber [114] for humidity sensing, which has a faster response time—3.5 s—and recovery time— 4 s—over commercial polyimide substitutes. By wrapping yarn as a dielectric layer on copper wire electrodes, a biaxial-sheathed shape is fabricated based on the outstanding humidity-transmission ability of yarns. As water molecules transport on fiber and yarn sensors, the structural design contributes to the larger surface area of each fiber, and the cross-shaped connection further enhances the ability to hold more water molecules in the sensor and form directional water movement on fibers. Similar designs include utilizing hydrophilic groups of amino acids in silk fiber [142]. Wang et al. constructed fiber-based porous membranes [117] inspired by water transportation in vascular plants. They generated nanofibers through electrospinning, then used dip-coating and electrospraying methods to fabricate the biomimetic membrane. Such membranes have ultrahigh one-way moisture transport capability and hold an outstanding water evaporation rate 5.8 times higher than cotton fabric.

Recent rapid developments in textile-based triboelectric and piezoelectric nanogenerators (NGs) will inevitably boost next-generation intelligent wearable electronics [143], as Figure 7b shows. A textile-based NG combines both mechanical energy harvesting and sensing abilities, whilst providing the versatile design of a carrier from the flexible platform of textiles. Current difficulties include the trade-off between the achievements of outstanding electrical performance and textile properties in fiber-based NGs [144].

The wide use of fibers in many other fields have been applied to a number of practical applications, for example, fibrous porous media. Model design of fibrous porous media [145,146], such as nanofibrous and microfibrous channels for faster capillary flow, were also reported to improve the performance of fluidic devices. Furthermore, in addition to the above areas, another direction worth investigating is interpenetrating polymer networks (IPNs), which can bring about a multitude of smart solutions in polymer and fiber science. Distinguished from other multipolymer combinations, IPNs are stable in solvents with suppressed creep and flow [147] and are thus suitable for biomedical applications, damping materials, tough and impact-resistant materials, etc. Some examples include light-gated control [148], greener extraction [149], increased robustness [150] and organic photovoltaic inks [151].

Besides electronic polymer-based fibers, wearables using smart and intelligent polymer-based membranes are gaining rapid growing interest due to their precise adjustability, permeability and adaptive properties [152]. As Chu et al. reported [153], boronate crosslinking bonds provide excellent elasticity of break strength at 33.4 MPa and 17.8% elongation. Using bioinspired membranes with adaptable wettability [154], the on-demand smart separation of an oil–water mixture [155] can be achieved, according to Li et al. With a similar idea, Liu et al. developed a semipermeable capsule membrane system [156] to detect and trace lead ions by selectively letting lead ions and water pass through.

For environmental or biological monitoring, molecularly imprinted polymers are also actively used. For instance, a fluorescence surface imprinted sensor towards tetracycline (TC) detection was reported by Wang et al. [8]. When combined with conductive fabrics, as Yinben et al. reported [157], a hybrid triboelectric nanogenerator fabricated by silk fibroin and poly(vinylidene fluoride) possesses great power-density performance and electrical properties.

## 4. Outlook

To summarize, we have attempted to redefine the ‘intelligence’ of a polymer by deducting from the term’s use in artificial intelligence. When referring to an intelligent material, it is wise to evaluate the degree of intelligence as equivalent to ‘intelligence’ in humans’ neural networks, inspired by biological sensing-reacting-learning behaviors in nature. As an intelligent polymer, the material should be able to process external signals and, at the same time, deal with signals generated internally to ensure the capacity to accomplish its desired goals. For future studies, the authors are of the opinion that it is necessary to clarify the difference between intelligent polymers and smart polymers, given the ambiguous boundary in-between and the considerable occasions they substitute for each other without distinction. As a consequence of small environmental variations, smart polymers undergo large reversible changes, either physical or chemical, in their properties. Although stimuli-responsive features are considered to be the main characteristic of smart polymers, intelligence should be considered more than responsiveness, because it holds the key point: the capacity to learn from the past, which is the premise of comprehension, adaption and active decision-making for future cases. Therefore, with the above discussion, one could notice that a nonlinear scale could be used as a more reasonable standard to define the intelligence of a material.

To make use of real ‘intelligent’ polymers, further systematic research is required to bring about the potential, from composition to the structural design of different materials. Specifically, intelligent SMPs should be beyond merely responsive to environmental changes but rather be able to select from behaviors and perform an action, which requires it to firstly, be not limited to one memorized shape, and secondly, to react with respect to different conditions. For self-healing polymers, future works could emphasize the optimization of their mechanical properties and their compatibility with alloys or ceramics to open new avenues in multiresponsive and multifunctional systems for advanced applications. Multiresponsive polymers are inevitably heading the path to more comprehensive utilization in complex environments.

## Figures and Tables

**Figure 1 polymers-13-01427-f001:**
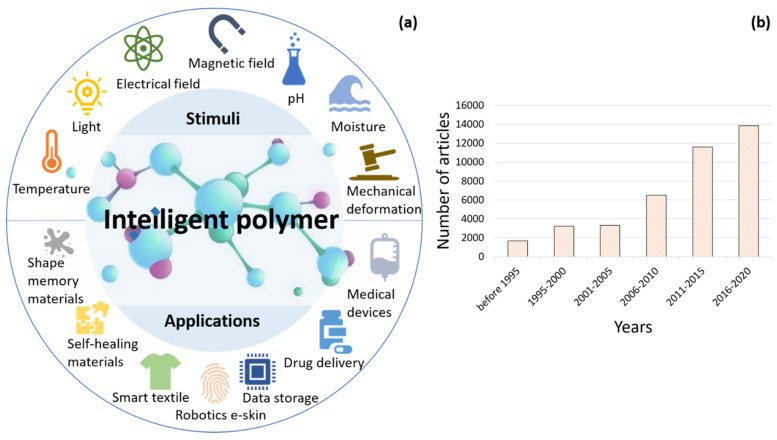
An overview of intelligent polymer development: (**a**) stimuli-responsive intelligent polymers and their diverse applications; (**b**) progress in intelligent polymer applications: an inclination in publications. Data derived from Scopus [24].

**Figure 2 polymers-13-01427-f002:**
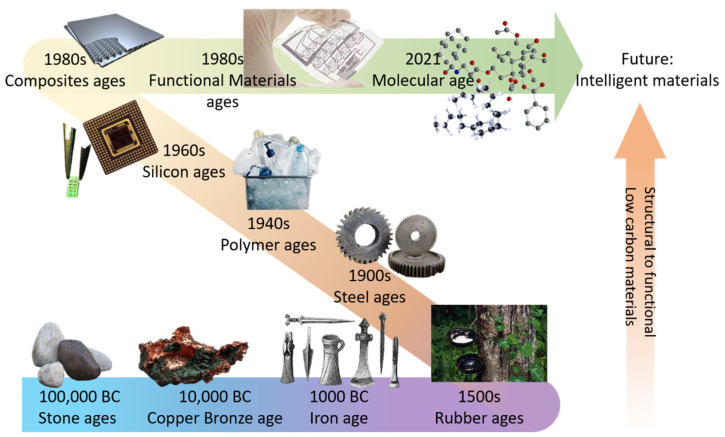
Historical and emerging tendencies of engineering materials.

**Figure 3 polymers-13-01427-f003:**
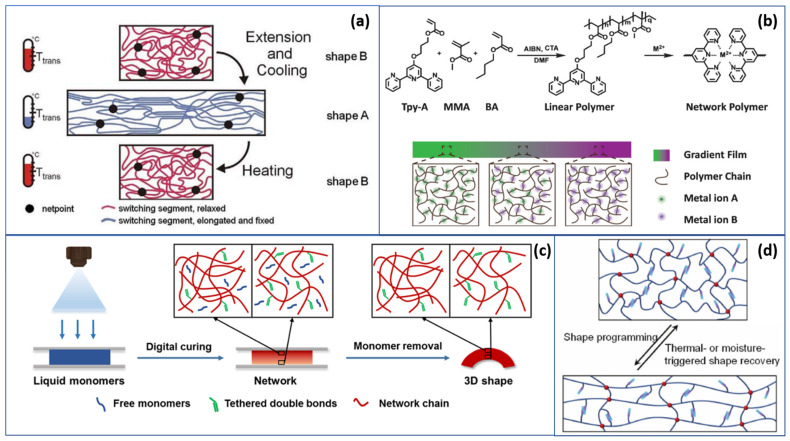
An overview on shape-memory intelligent polymers: (**a**) working mechanism of a thermoresponsive SMP [38,60]; (**b**) design and polymer-metal network synthesis [58]; (**c**) schematic illustration of the digital-light-printed SMP (Reprinted with permission from Zhang, Y.; Huang, L.; Song, H.; Ni, C.; Wu, J.; Zhao, Q.; Xie, T. 4D Printing of a Digital Shape Memory Polymer With Tunable High Performance. *ACS Appl. Mater. Interfaces* **2019**, *11* (35), 32408–32413, Copyright American Chemical Society) [54]; (**d**) thermal- and humidity-triggered response of zwitterionic SMP [61].

**Figure 4 polymers-13-01427-f004:**
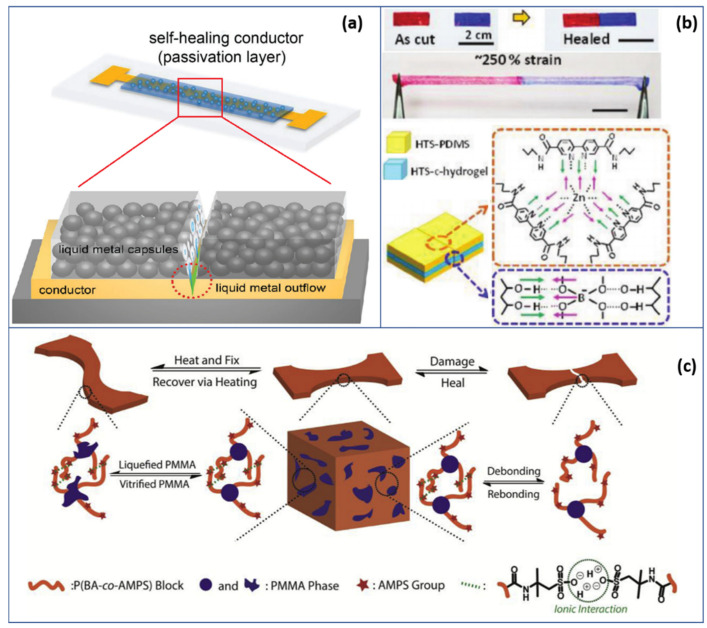
An overview on self-healing intelligent polymers: (**a**) self-healing mechanism of a metal conductor by liquid metal outflow from a polymer capsule [91]; (**b**) self-healing and stretching behavior of an HTS-PDMS gel under ambient conditions [95]; (**c**) self-healable thermoplastic material. The healing property relies on chain mobility and the rebonding of the molecular interactions (Reprinted from (Polymer, Volume 134, Zhang, J., Huo, M., Li, M., Li, T., Li, N., Zhou, J., Jiang, J., 2018. Shape memory and self-healing materials from supramolecular block polymers. Pages 35–43, 2018 with permission from Elsevier) [90].

**Figure 6 polymers-13-01427-f006:**
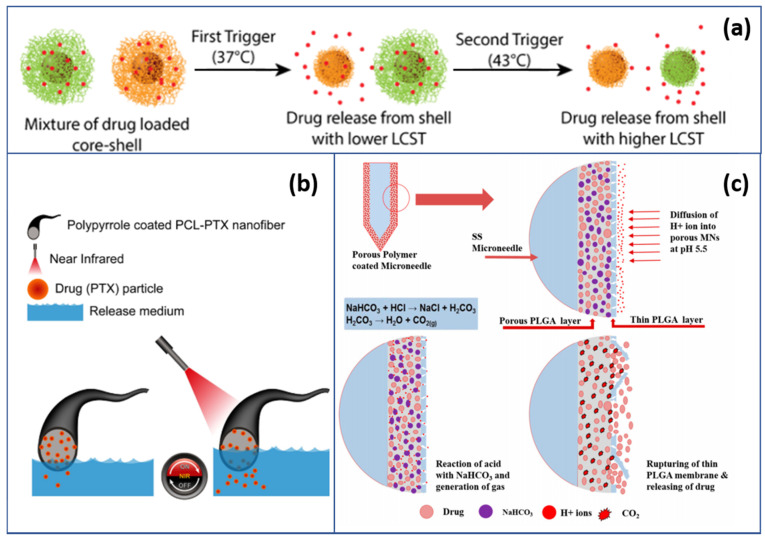
Applications using responsive polymers: (**a**) schematic representation of the trigger-dependent sustained release of a drug from a mixture of nanoparticles having varying trigger sensitivities [121,122]; (**b**) schematic illustration of the NIR-triggered drug release from the PPy-coated fiber (Reprinted with permission from Tiwari, A. P.; Hwang, T. I.; Oh, J.-M.; Maharjan, B.; Chun, S.; Kim, B. S.; Joshi, M. K. K.; Park, C. H.; Kim, C. S. pH/NIR-Responsive Polypyrrole-Functionalized Fibrous Localized Drug-Delivery Platform for Synergistic Cancer Therapy. *ACS Appl. Mater. Interfaces*, **2018**, *10*(24), 20256–20270. Copyright American Chemical Society) [118]; (**c**) schematic illustration of the pH-responsive release system [123].

**Figure 7 polymers-13-01427-f007:**
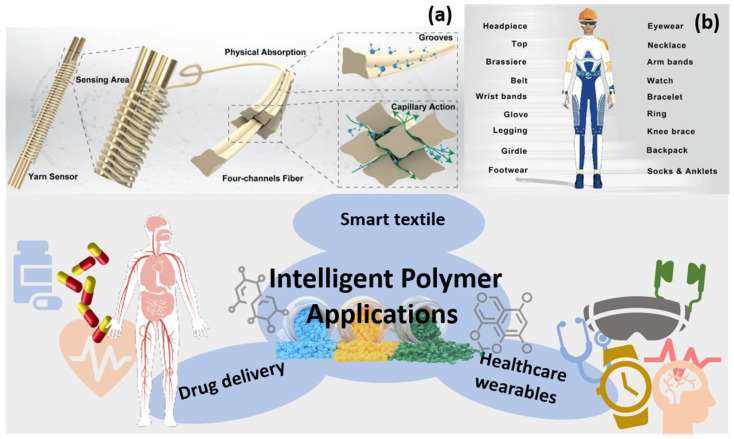
Trending topics in intelligent polymer applications: smart textile, drug delivery and healthcare wearables; (**a**) full-textile wireless flexible humidity sensor for human physiological monitoring [114]; (**b**) smart textile assemblies in wearable microelectronic systems [136].

**Table 1 polymers-13-01427-t001:** Performance of various shape-memory polymers.

Material	Feature	Shape Fixity	Shape Recovery Rate	Method	Application	Ref.
MA/IBOA/HDDA	Shapeshifting through time	100%	>97.2%	Digital light curing	Nanophotonics, shrinkable electronics	[54]
tBA/HDDA/TPO	Photosensitive	96%	100%	Digital light processing printer	Aerospace domain and biomedical applications	[62]
Tpy-A/MMA/BA/DMF	Plastic morphing versatility	99%	95%	Chemical synthesis	3D shape construction	[58]
PU/Al foil	Triboelectric output when recovering	…	>95%	Electrospinning	Self-powered wearable devices	[63]
Polycaprolactone/Ebecryl 8413/BA	Color changing feature, thermal triggered	97%	97%	Chemical synthesis	Soft robotics, artificial skins	[64]
SA/P(AA-AM)/LiCl Hydrogel	Controllable thermal-responsive	95%	97%	Radical polymerization	Motion-sensing element	[65]

**Table 2 polymers-13-01427-t002:** Comparison of selected novel intelligent polymers’ responses to mechanical deformation stimulus and their performance.

Material	Novelty	Fabrication Method	Working Mechanism	Shape Recovery Ratio	Strain Resistance	Tensile Strength	Durability	Application	Ref.
(PMMA-b-P(BA-co-AMPS))	Self-healable shape-recovery ability	RAFT copolymerization	Physical supramolecular crosslinking /interactions	95%	500%	10 MPa	…	Nanophotonics, shrinkable electronics	[90]
PU-co-TPGDA	Shape recovery no longer relies on thermoprogramming; instantaneous shape recovery	Colloidal templating	Changes in solvability of swelling polymer solvent triggered by evaporation	100%	…	…	>500 cycles	Aerospace domain and biomedical applications	[96]
Graphene-rubber elastomer nanocomposite	Sensing strain with high resolution of 0.125%	Chemical synthesis followed by vacuum filtration process	Thermoelectricity	…	200%	Sensitivity ~2.52 ln(nA)/%	>1000 cycles	3D-shape construction	[34]
CNF−PPy/PB hybrid hydrogel	Low density and biocompatible self-healing hydrogel	Polymerization	Hierarchically conductive network	100%	600%	∼62.8 kPa	>1500 cycles	Motion-sensing element	[97]

**Table 3 polymers-13-01427-t003:** Selected comparison of multiple stimuli-driven polymer applications.

Application	Category	Material	Fabrication Method	Working Mechanism	Features	Novelty	References
Moisture-wicking fabric	Moisture-sensitive polymer	C6FPU	Electrospray, dip-coating self-synthesized	Water transport due to differential capillary forces	Water evaporation rate of 0.67 g/h	Biomimetic membrane fastens water evaporation and transportation	[117]
Humidity-driven actuator	Moisture-sensitive polymer	CNT-paper/BOPP	Dip-coating of CNT-paper composite, attached with BOPP film	Hygroexpansion effect	Curvature change from 1.2 cm-1 to 0 w.r.t relative humidity change 14% to 60%	Dual-mode actuating performance	[111]
Shape-programmable soft robotics	Moisture-sensitive polymer	PVDF/APTES	Spin-coating with chemical synthesized template	Hygroscopicity of microchanneled film	Bending and coiling due to anisotropic flexural modulus	Microchannel structure design on one side of polymer film	[110]
Piezo-switchable surface	Electric field triggered	PVDF/PMMA	Electrospinning	Functional groups rearrangement	Fabricated surface responds to the electric field	Tunable surface water/oil wettability	[113]
Site-specific drug delivery	pH-/heat-responsive	Polypyrrole-coated PCL-PTX	Electrospinning	Physicochemical characteristics of fibrous mats	Superior drug release in environment with pH 5.5	Stepwise-based drug-release behavior	[118]
Corrosion sensing and protection	pH-sensitive	HQ/HQSEA	Chemical synthesis	Release of encapsulated corrosion inhibitor under acidic condition	Enhanced fluorescence on the material after nanoparticles released	Corrosion sensing and protection	[119]
Wearable sensing devices	pH-/humidity-sensitive	polyPtC	Chemical synthesis	Crystalline−amorphous transition	Reversible yellow-to-black electrochromism	Structural design of metallo-supramolecular polymer	[116]

## Data Availability

Not applicable.

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
