# Peer review of "Intelligent Polymers, Fibers and Applications"

_polymers, 2021, doi:10.3390/polym13091427_

Round 1

Reviewer 1 Report

Depending on the way of molecules bonded with each other, the unique properties of polymers are gaining increasing attention and investigations. The characteristics of different polymers rely on how the long chains of the molecules repeat themselves. Intelligent polymers, as the word 'intelligent' named itself, are referring to those polymers responsive to single or multiple stimuli, which could be either chemical or physical. Intelligent materials, also known as smart materials, are capable of reacting to various external stimuli or environmental changes by rearranging their structure in molecular-level and adapting functionality accordingly. In this paper, the most recent progress of smart materials with various features are reviewed with focus on their applications in diverse fields. Moreover, their performance and working mechanisms based on different physical, chemical, and biological stimuli, such as temperature, electric and magnetic field, deformation, pH and enzymes etc. are summarized. Although the topic in this work was interesting, the presentation in this manuscript was very poor. This manuscript should be rejected for published in Polymers. However, if the authors are willing to make the substantial revisions according to my comments, I would be glad to re-review this manuscript. Here are my detailed comments:

  1. The detailed literature review indicates efforts made by the authors. The coherence of the related work, however, is still not clear. It may help the authors by answering the following questions: Why are these works relevant? Which specific problems were addressed? How are the previous results related with the latest work? What are the outstanding, unresolved, research issues? Which of them has been solved by the proposed study? Answering the questions leads to the novelty of the proposed work naturally. Besides, the current one is nothing but a literature review. Why their work is important comparing to previous reports? I think this is essential to keep the interest of the reader.
  2. In Table 1 and 2, the authors should give the explanations for the difference of data collected from different sources.
  3. However, further effort is still required in the design and synthesis of all-organic magneto-responsive materials that will allow them to enter new unexplored fields. The findings and their implications should be discussed in the broadest context possible. Future research directions may also be highlighted. The authors should give some explanation on above conclusions. What the authors meant is not clear to reader.
  4. Although the results look “making sense”, the authors should dig deeper in the results by presenting some in-depth discussion, such as implications of the results, such as possible application of them.
  5. Fibers have been widely used in many fields of life. Fibers have been applied in a number of practical applications, for example fibrous porous media, (see [A fractal model for capillary flow through a single tortuous capillary with roughened surfaces in fibrous porous media, Fractals, 2021, 29(1):2150017; Fractals, 2019, 27(7): 1950116]). Authors should introduce some related knowledge to readers. I think this is essential to keep the interest of the reader.
  6. Please, expand the conclusions in relation to the specific goals and the future work.

Author Response

Please see the attachment. Thank you :)

Reviewer 2 Report

The manuscript related to intelligent polymers, fibers, and applications has been reviewed. The manuscript is very well written and well presented.
I appreciate authors summarizing such a broad field carefully into one review paper. 
According to my thinking, the author might have missed one of the essential emerging classes of intelligent polymers, i.e., vitrimers; the authors suggested reading and cite some classical papers. This important class is now biobased and offers various functionalities, self-healing, self-welding, and others, as described in vanillin-based epoxy vitrimer with high performance and closed-loop recyclability and other papers.

Author Response

Please see the attachment. Thank you :)

Reviewer 3 Report

The review by Jingcheng and co-workers details the interesting field of intelligent polymers/materials. The review is detailed and most of the relevant works are mentioned. However, there are specific areas that need to be included for comprehensiveness. Critical discussions needs to be added as well. The following comments should be addressed before further consideration.

1. How was panel c) in Figure 1 derived? The three columns look more or less the same, and there is no clear trend. This panel should be deleted as it brings up too many debatable questions.

2. The authors should add more critical discussions along the introduction of each section to turn the text into a review instead of merely listing the different works. Critical assessment of the literature is crucial and should be deepened.

3. Almost all the figures were taken from the literature directly. The authors should construct some figures that represent concepts and comparisons in innovative ways, and not just reproduce the illustrations from the literature. The comparison tables are useful but the authors could go beyond that.

4. Some additional examples of polymers that give different performance based on the pH should be briefly mentioned in the review (POM 10.1021/acsanm.0c02365; PBI/PIM 10.1021/acsanm.8b01563; GCP 10.1021/acsapm.0c00394; PtC 10.1021/acsapm.0c00782).

5. Novel nanosized biomass-based pH-responsive cellulose nanofibers should be mentioned in the review (10.1021/acs.jafc.9b06588).

6. The authors start the review by introducing intelligence, and artificial intelligence in a very general sense. This is useful to the readers but some examples of their use in polymer/fiber/material preparation should be mentioned, which goes even beyond conventional smart materials; examples could include defected materials (10.1021/acsnano.0c05267), sustainable materials (10.1039/D0GC02956D), polymers (10.1021/acsapm.0c00586).

7. Intelligent wearable electronic fibers should also be at least mentioned in the review (10.1002/adma.201902549; 10.1002/adsu.202000108).

8. Smart and intelligent polymer-based membrane systems should also be highlighted in an individual section/paragraph as the area is growing rapidly (10.1016/j.memsci.2020.118912; 10.1016/j.memsci.2020.118523; 10.1016/j.memsci.2019.117661; 10.1016/j.memsci.2020.118439; 10.1016/j.memsci.2020.117954; 10.1016/j.memsci.2020.118304).

9. Interpenetrating polymer networks should be also discussed in a paragraph as it can bring about a multitude of smart solutions in polymer and fiber science. Some examples include light gated control (10.1002/adfm.202005328), greener extraction (10.1039/D1GC00148E), increased robustness (10.1021/acsnano.8b04123), inks (10.1039/C9GC02288K).

10. The outlook section should focus more on the current research trends, and where the researchers should place the emphasis. The authors should give some directions that helps the community to identify the most challenging problems and sought-after possible solutions in the field.

Author Response

Please see the attachment. Thank you :)

Round 2

Reviewer 1 Report

It is ok.

Reviewer 2 Report

The authors have made necessary changes, the manuscript might be considered for publication.

Reviewer 3 Report

The authors have addressed the comments and the manuscript improved.